# Energy-Saving One-Step Pre-Treatment Using an Activated Sodium Percarbonate System and Its Bleaching Mechanism for Cotton Fabric

**DOI:** 10.3390/ma15175849

**Published:** 2022-08-25

**Authors:** Qing Li, Run Lu, Yan Liang, Kang Gao, Huiyu Jiang

**Affiliations:** 1Hubei Key Laboratory of Biomass Fibers and Eco-Dyeing & Finishing, College of Chemistry and Chemical Engineering, Wuhan Textile University, Wuhan 430200, China; 2Jiangsu Engineering Research Center of Textile Dyeing and Printing for Energy Conservation, Discharge Reduction and Cleaner Production, Soochow University, Suzhou 215123, China; 3Key Laboratory of Clean Dyeing and Finishing Technology of Zhejiang Province, Shaoxing University, Shaoxing 312000, China; 4National Innovation Center of Advanced Dyeing and Finishing Technology, Tai’an 271000, China

**Keywords:** pre-treatment, cotton, sodium percarbonate, bleaching mechanism, energy conservation

## Abstract

The traditional pre-treatment of cotton fabric hardly meets the requirement of low carbon emissions due to its large energy consumption and wastewater discharge. In this study, a low-temperature and near-neutral strategy was designed by establishing a tetraacetylethylenediamine (TAED)-activated sodium percarbonate (SPC) system. First, the effects of SPC concentration, temperature and duration on the whiteness index (*WI*) and capillary effect of cotton fabrics were investigated. Particularly, excess SPC’s ability to create an additional bleaching effect was studied. The optimized activated pre-treatment was compared with the traditional pre-treatment in terms of the bleaching effect and energy consumption. Further, the degradation of morin, which is one of the natural pigments in cotton, was carried out in a homogeneous TAED/SPC system to reveal the bleaching mechanism. Lastly, the application performance of the treated cotton was evaluated by characterizing the dyeability, mechanical properties, morphology, etc. The research results showed that temperature had a significant influence on both the *WI* and capillary effect, followed by the SPC concentration and duration. The *WI* was positively correlated with the SPC concentration, but excess SPC could not produce an obvious additional effect. The *WI* of the fabric increased by 67.6% after the optimized activated bleaching using 10 mmol/L SPC and 15 mmol/L TAED at 70 °C for 30 min. Compared with the traditional process performed at 95 °C for 45 min, the activated process produced approximately 39.3% energy savings. Research on the bleaching mechanism indicated that the reactive species that participated in degrading the morin were the hydroxyl radical and superoxide radical, and the contribution degree of the former was larger than that of the latter. Two degradation components with molecular weights of 180 and 154 were detected using mass spectroscopy. Based on this, the bleaching mechanism of the TAED/SPC system was proposed. Moreover, the fabric after the activated pre-treatment had a suitable dyeability and strength, a lower wax residual and a smoother and cleaner fiber surface. The encouraging results showed that TAED/SPC is a promising bleaching system that is conducive to the sustainable advance of the textile industry.

## 1. Introduction

Recently, society’s increasingly earnest pursuit of carbon neutrality is driving the textile industry to urgently pursue sustainable development. The carbon emissions from the textile industry are 8–10% of the world’s emissions, which directly aggravates climate change [1]. Conventional pre-treatment for textiles involves sewage discharge and high-energy-consuming processes, which hardly meet the sustainability criteria [2]. However, pre-treatment is an essential step before dying and finishing to remove the natural and artificial impurities from the fiber, thereby giving the textile a white appearance and good hygroscopicity. Therefore, searching for a low-temperature and near-neutral technology strategy for pre-treatment to decrease the consumption of energy and disposal of effluent is beneficial to realizing the goal of carbon neutrality.

Cotton fiber, which is one of the important natural cellulose materials, displays many advantages, such as soft handling, fine breathable moisture absorption and excellent dyeability. Not only that, its easier biodegradation in nature avoids the potential risks of contamination caused by microfibers in underground water, lakes and ocean [3]. Thus, the textile industry and consumer market have begun to pay attention to cotton fiber again. Generally, mature cotton contains some non-cellulosic impurities (e.g., wax, pectin, ash, pigment, etc.), which significantly impair the water absorbency and whiteness of the fiber [4]. The traditional pre-treatment, which mainly includes scouring and bleaching, is performed using hydrogen peroxide (H_2_O_2_) at approximately boiling temperature and in a medium containing an alkali and a surfactant [5]. H_2_O_2_ ionizes and generates perhyroxyl anions (HO_2_^−^), which attack the pigment impurity through a nucleophilic reaction to improve the *WI* of the fabric [6]. Such drastic conditions cause a large amount of energy consumption, and the disposal of alkaline effluent is difficult due to its high chemical oxygen demand (COD) [7]. More seriously, the cotton fiber probably undergoes potential damage owing to the excessive oxidation of the molecular chain of cellulose; therefore, the fabric strength will decrease significantly during the subsequent process, especially in alkaline environments.

It was confirmed that adding an H_2_O_2_ activator in a traditional pre-treatment solution is a convenient and effective method to obtain a low-temperature and near-neutral process. An H_2_O_2_ activator is a kind of compound that can react with H_2_O_2_ to generate peracids in situ [8]. Peracids are able to oxidize and degrade the pigment impurity in cellulosic fibers at a lower temperature due to the higher redox potential compared with H_2_O_2_ [9,10]. After the oxidation of the pigment, peracids can easily decompose and generate acid compounds that can neutralize part of the alkali, thus providing a near-neutral processing environment [11]. Although various H_2_O_2_ activators were extensively researched, tetraacetylethylenediamine (TAED) is the only industrialized activator that shows the advantages of being non-toxic, non-sensitizing and biodegradable [12,13,14,15,16]. The much lower toxicity of TAED to aquatic animals compared with other activators was confirmed in a recent report [13].

A TAED-activated H_2_O_2_ system was established to bleach cotton, which was confirmed to function effectively [17,18]. Regarding the TAED/H_2_O_2_ system, the decomposition rate of H_2_O_2_ is a significant factor for achieving the desired bleaching result because having decomposition that is too quick may generate large amounts of reactive bleaching species that are unable to degrade pigment due to their short life span and are even harmful to the fiber strength [5]. Furthermore, liquid H_2_O_2_ and solid TAED granules should be placed separately and employed immediately after mixing, resulting in an inconvenient application during the practical processing.

Sodium percarbonate (SPC, 2Na_2_CO_3_·3H_2_O_2_) is an addition compound of sodium carbonate (Na_2_CO_3_) and H_2_O_2_. SPC has been widely applied as household and industrial detergents due to its non-polluting property [14,19,20]. There are also some reports about SPC bleaching that focused on mechanical pulp [21,22], printed paper [23], lining leathers [24], nonvital teeth [25] and so on. After being dissolved in water, SPC can steadily release H_2_O_2_ to keep a suitable decomposition rate of H_2_O_2_, thus reducing thus reducing the invalid decomposition of H_2_O_2_ [6]. The reaction between H_2_O_2_ and TAED is promoted under slightly alkaline conditions [26], and thus, the Na_2_CO_3_ released by SPC can drive the reaction to completion without adding additional alkaline. Meanwhile, it is feasible to premix the solid SPC with TAED and deposit them for a future application. Therefore, in this study, an SPC/TAED system instead of a H_2_O_2_/TAED system was established to improve the performance and convenience of bleaching cotton fabric.

Usually, the poor solubility of TAED limits its bleaching application at a lower temperature, such as 50 °C. Our recent research confirmed that assisting the process with ultrasound accelerates the dissolution of TAED and remarkably promotes the bleaching performance for cotton/spandex fabric [27]. However, the introduction of ultrasound increases the input cost of equipment. The cost of TAED (6000 USD/ton) is much higher than that of SPC (800 USD/ton) (data from Zhejiang Jinke Chemicals Co., Ltd., Shaoxing, China). To decrease the cost input of equipment and chemical reagents without weakening the bleaching result, it is necessary to explore the influence of SPC concentration on the bleaching and scouring effects of cotton fabric using a conventional water bath rather than ultrasound. More specifically, excess SPC was added to create a TAED/SPC system to investigate whether there was an additional bleaching effect. It was speculated that some of the SPC reacts with TAED and produces a peracid, which combines with the excess SPC to create more reactive bleaching species that are responsible for degrading pigments. Moreover, the temperature and time were also analyzed to obtain an optimized pre-treatment process. The low-temperature process was further compared with the traditional process in terms of the whiteness index and hygroscopicity of cotton fabric, pH and energy consumption of the process.

The chemical reactions that take place in a TAED/SPC bleaching system are displayed in Figure 1. One molecule of SPC decomposes with the release of three molecules of H_2_O_2_ [14]. One molecule of TAED can react with two molecules of H_2_O_2_ to generate two molecules of peracetic acid (PAA) [17,28,29]. PAA is a more kinetically active bleaching alternative compared with H_2_O_2_ [4,30]. The hydroxyl radical (HO·) as the decomposition product of PAA is also considered an efficient bleaching agent [27,31]. Moreover, singlet oxygen (^1^O_2_) can be generated from the bimolecular decomposition of PAA but the bleaching effectiveness is weak, probably due to its short lifetime [26,32]. However, no special research has been carried out to explore the reactive oxygen species (ROS) that play a role in TAED/SPC bleaching. Furthermore, the contribution degree of different ROS to the degradation of pigment in cotton is unclear. To investigate the above issues, morin as one of the natural pigments with flavonoid structure in mature cotton fiber was selected in this research [33]. The effects of different radical scavengers on the degradation rate of morin in a TAED/SPC system were tested to find effective ROS for bleaching. The degradation products were analyzed using liquid chromatography–mass spectrometry (LC-MS), mass spectroscopy (MS) and hydrogen nuclear magnetic resonance spectroscopy (^1^H-NMR) to speculate the degradation pathway of morin. Finally, the mechanical properties and dyeability of the bleached fabric using TAED/SPC were evaluated. The morphology and surface functional groups of the bleached fabric were observed using scanning electron microscopy (SEM) and attenuated total reflection Fourier-transform infrared spectroscopy (ATR FT-IR), respectively.

## 2. Materials and Methods

### 2.1. Materials

Desized, unscoured and unbleached pure cotton woven fabric with a density of 210 g/m^2^ and a whiteness index of 40.10 was selected and used for the pre-treatment. TAED (cream-colored granules, purity 91.3%) and SPC (granules with a stabilizing coating, content of active oxygen 13.8%) were the bleaching auxiliaries and were bought from Zhejiang Jinke Chemicals Co., Ltd. The H_2_O_2_ (purity 30% *w*/*w*), sodium carbonate (Na_2_CO_3_), sodium hydroxide (NaOH) and sodium sulfate (Na_2_SO_4_) were of analytical reagent grade. The non-ionic penetrating agent and neutral detergent were of industrial reagent grade. Reactive dyes with three primary colors applied in the dyeing process were obtained from Hubei Colour Root Technology Co., Ltd., Jingzhou, China. Morin (purity 95%), nitrotetrazolium blue chloride (NBT, purity 98%) and radical scavengers, such as dimethyl sulphoxide (DMSO), benzoquinone (BQ) and sodium azide (NaN_3_), were purchased from Aladdin. Except for the washing and rinsing of the fabric, deionized water was used for all experiments.

### 2.2. One-Step Pre-Treatment

The pre-treatment based on the activated SPC system was performed using a universal water bath dyeing machine. The weight of each fabric was 4 g. The weight ratio of the fabric to treating fluid was 1:25, enabling the fabric to be soaked completely and obtain a uniform treatment effect. The fabric samples were completely immersed in the treating solutions containing SPC, TAED and the penetrating agent. The temperature of the solution was increased to the required value and kept for a certain time. Lastly, the fabric was washed with running water for 3 min and then allowed to air-dry.

For comparison purposes, the conventional process was carried out using the treating solution containing 30–800 mmol/L H_2_O_2_ (30% *w*/*w*), 2 g/L Na_2_CO_3_ and 2 g/L penetrating agent at 95 °C for 45 min [34]. After that, the fabric sample was treated as mentioned earlier. To diminish the experimental error, each process was performed three times in parallel. 

### 2.3. Morin Degradation Experiment

Morin degradation in the TAED/SPC homogeneous system was performed to simulate the bleaching of natural pigments in cotton fiber. First, morin was completely dissolved in the water by adding NaOH. Then, the morin solution was mixed with the TAED/SPC bleaching solution and immediately heated to the required temperature. The final mixed solutions were monitored using a TU-1950 UV-Vis spectrophotometer (Beijing Persee General Instrument Co., Ltd., Beijing, China) at different time points. 

### 2.4. Radical Scavenger Experiment

To detect the active radical species that degrade morin in the TAED/SPC system, a radical scavenging experiment using dimethyl sulphoxide (DMSO), benzoquinone (BQ) and sodium azide (NaN_3_) was carried out. Nitrotetrazolium blue chloride (NBT) was also applied to probe the radical. To verify the impact of valid active radicals on the *WI* of the bleached fabric, DMSO and BQ were separately added to the actual bleaching solution to scavenge radicals. The fabric was bleached at 70 °C for 30 min by immersing it in a solution containing 2 g/L penetrating agent, 15 mmol/L TAED, 10 mmol/L SPC and 0–4 mL DMSO or 0–0.025 g BQ.

### 2.5. Dyeing with Reactive Dyes

The activated bleached fabric was treated using 10 mmol/L SPC and 15 mmol/L TAED at 70 °C for 30 min. The conventional bleached fabric was treated using 100 mmol/L 30% H_2_O_2_ at 95 °C for 45 min. A competing dyeing process was carried out by simultaneously immersing the activated and conventional bleached fabrics in the same dyeing bath. The dyeing bath was heated to 60 °C at a rate of 3 °C/min and kept for 50 min. During the holding stage, 15 g/L Na_2_SO_4_ and 15 g/L Na_2_CO_3_ were respectively added into the dyeing bath at the 5th min and 20th min marks. At the end of the dyeing process, the dyed fabrics were removed, washed with the neutral detergent to remove the unfixed dye and washed with running water to avoid an uneven apparent color depth.

### 2.6. Analytical Methods

#### 2.6.1. Color Features

Color features of the fabric, i.e., the CIE *L*, CIE *a*, CIE *b*, tristimulus values (*X*, *Y* and *Z*), color depth (*K/S*) and whiteness index (*WI*) of the fabric sample, were tested with a 110 reflectance spectrophotometer (Datacolor, Lawrenceville, NJ, USA), adopting illuminant D_65_ and a 10° standard observer. The yellowness index (*YI*) and chromatism (Δ*E*) were calculated using Equations (1) and (2), respectively [28,35]:*YI* = 100 × (1 − 0.847 × *Z*/*Y*)(1)
Δ*E* = [Δ*L*^2^ + Δ*a*^2^ + Δ*b*^2^]^1/2^(2)
where Δ*L*, Δ*a* and Δ*b* are the differences in CIE *L*, CIE *a* and CIE *b* between the two dyed fabric samples, and Δ*E* is used as the quantitative assessment of the color differences between two dyed samples. 

#### 2.6.2. Hygroscopic Property

This test was carried out according to FZ/T01071-2008, which references the ISO standard 9073-6:2000. The width and height of the fabric sample were 3 cm and 25 cm, respectively. The above fabric sample was stabilized on a steel stand vertically. The bottom end (15 mm) of the fabric was immersed below the water surface. The height of water rising along the fabric within 30 min demonstrates the capillary effect. A higher capillary effect indicates a better hygroscopic property.

#### 2.6.3. Mechanical Properties

The mechanical properties of the fabric were tested according to GB/T3916-1997 <Measurement of breaking strength and elongation at break for textiles, yarn from packages and single yarn>, which is equivalent to ISO 2062:1993. The universal material testing machine was employed for the test with a clip distance of 20 cm, tension speed of 20 cm/min and applied load range of 0–2500 N. The size of the sample was 50 mm × 250 mm. Before the test, the sample was placed in a closed space with a constant temperature and humidity (25 °C, 65%) for 12 h. An average value of each sample was calculated after 5 tests in parallel. The activated bleached sample was treated using 10 mmol/L SPC and 15 mmol/L TAED at 70 °C for 30 min. The conventional bleached sample was treated using 100 mmol/L 30% H_2_O_2_ at 95 °C for 45 min.

#### 2.6.4. SEM and ATR FT-IR

The micromorphology of the fabric was observed using a scanning electron microscope (Phenom, The Netherlands). The test sample was fixed on a stub using conducting adhesive tape and uniformly coated with gold. An activated bleached sample treated using 10 mmol/L SPC and 15 mmol/L TAED at 70 °C for 30 min was used. A conventional bleached sample treated using 100 mmol/L 30% H_2_O_2_ at 95 °C for 45 min was used. 

ATR FT-IR spectra (Tianjin Gangdong, Tianjin, China) were collected with a universal ATR accessory (Lambda Scientific Pty Ltd., Miami, FL, USA) within the wavelength range of 750–4000 cm^−1^. A 20 mm × 20 mm piece of each sample was prepared and pressed on top of an ATR cell. The OPUS software (Version 8.7) (Stuttgart, Germany), which is the specialized software for the measuring, processing and assessment of infrared spectroscopy, was adopted in this test. The activated bleached sample treated using 10 mmol/L SPC and 15 mmol/L TAED at 70 °C for 30 min was used. A conventional bleached sample treated using 100 mmol/L 30% H_2_O_2_ at 95 °C for 45 min was used.

#### 2.6.5. UV-Vis Spectroscopy

The UV-Vis absorption spectra of morin solutions with different concentrations (10–100 μmol/L) were measured and the results are shown in Appendix A. The characteristic absorption peaks at 275 nm, 317 nm and 410 nm correspond to the benzoyl group, cinnamic acyl group and conjugated double bond, respectively [36]. A good linear relationship between the absorbance (Abs) at the adsorption maximum wavelength (λmax, 410 nm) and the morin concentration is displayed in Appendix A. A satisfactory correlation R^2^ up to 0.9998 indicated the high credibility of the calculation of the morin concentration. Based on this fact, the morin concentration in the TAED/SPC bleaching solution prepared according to Section 2.3 at different time points during the degradation was estimated accordingly. Further, the retention rate of morin was expressed as the ratio of *C*/*C_0_*. Therein, *C* and *C_0_* are the morin concentrations at a certain time point and the initial time point during the degradation process, respectively.

#### 2.6.6. LC-MS, MS and ^1^H-NMR

The morin before and after degradation was detected using liquid chromatography–mass spectrometry (LCMS-8050, Shimazu) to determine whether the degradation reaction was complete. The mobile phase was a mixture of 0.025% ammonia water (A) and acetonitrile (B) (A:B = 95%:5%~5%:95%). The morin (6 mmol/L) was degraded in a homogeneous bleaching solution that included 50 mmol/L TAED and 34 mmol/L SPC at 70 °C for 1 h. The reaction liquid after degradation was frozen and dried at −60 °C for 16 h using a freeze dryer. The degradation product was a white powder and purified via column chromatography filled with silica using an eluent composed of methanol and methylene chloride. Further, the structure of the purified degradation product was tested using mass spectroscopy (MS, Shimazu, Kyoto, Japan) and nuclear magnetic resonance hydrogen spectroscopy (CD_3_OD, 400 Hz, ^1^H-NMR, Bruker-400, Billerica, MA, USA) to speculate on the degradation pathway of morin. 

## 3. Results

### 3.1. Analysis of Process Factor

#### 3.1.1. Effect of SPC Concentration

The fabric was simultaneously scoured and bleached in the solutions containing 2 g/L penetrating agent, 15 mmol/L TAED and 0–50 mmol/L SPC at 70 °C for 30 min. According to the molar ratio of the reaction between SPC and TAED, it was considered that 10 mmol/L was the stoichiometric amount of SPC relative to 15 mmol/L TAED. 

As depicted in Figure 2a, the *WI* increased rapidly with the increasing SPC concentration in the range of 0–10 mmol/L. This result was to be expected because of the gradually increasing PAA as the main bleaching oxidizer. However, the *WI* decreased slightly when the SPC concentration exceeded its stoichiometric amount, i.e., 10–30 mmol/L. To analyze this phenomenon, pH variations in the bleaching solution before and after bleaching were measured and are shown in Figure 2b. After the bleaching, the pH value in the range of 7–10 probably accelerated the decomposition of PAA [37], thereby producing a large amount of reactive oxygen species (ROS) in a short time. Some of the ROS took no part in the bleaching due to the quenching before the oxidation with the pigment. Interestingly, the *WI* was slightly promoted instead when the SPC concentration attained 30–50 mmol/L, which drastically surpassed its stoichiometric amount. The decrease in *YI* also verified this phenomenon. At this time, the solution was under strong alkali conditions, as shown in Figure 2b. The excess SPC released abundant H_2_O_2_, which ionized with the generation of HO_2_^−^ in this medium [26]. HO_2_^−^ was confirmed as a kind of effective bleaching species to improve the *WI*, which could explain why the *WI* increased instead. Moreover, it is worth mentioning that the *WI* of pristine cotton was increased from 40.10 to 49.23, even in the absence of SPC, indicating that the removal of other hydrophobic impurities was also conducive to the improvement in fabric appearance.

The water absorbency of the fabric was evaluated by testing the capillary effect. The capillary effect of woven fabric that reaches 10 cm/30 min or above indicates qualified water absorbency [38]. As Figure 2c illustrates, the capillary effect reached up to about 10 cm/30 min even without adding SPC, and no matter how much SPC was added, the capillary effect remained in the range of 10–12 cm/30 min. Therefore, it can be deemed that the SPC had little impact on the scouring of cotton. The capillary effect of pristine cotton was only 0.5 cm/30 min but the capillary effect of all the treated cotton was significant. The satisfactory water absorbency was mainly attributed to the penetrating agent used, which played a prominent role in removing the hydrophobic impurities. With an SPC concentration of 10 mmol/L, i.e., its stoichiometric amount relative to TAED, the highest *WI* was obtained; thus, this dosage was adopted to investigate the effect of temperature.

#### 3.1.2. Effect of Temperature

The fabrics were scoured and bleached at different temperatures (20–90 °C) in the solution containing 10 mmol/L SPC, 15 mmol/L TAED and 2 g/L penetrating agent for 30 min. As Figure 3a shows, the gradually increasing temperature contributed to a visible increase in *WI* and decrease in *YI*. This phenomenon may be interpreted as follows. Increasing the temperature gave direct energy to accelerate the release of H_2_O_2_ from the SPC, drive the reaction between H_2_O_2_ and TAED to completion, provide adequate reactive oxygen species (ROS) from the decomposition of H_2_O_2_ and PAA, enhance the accessibility between ROS and pigment impurities in cotton, and facilitate the oxidative degradation of pigments by ROS. 

Figure 3b reveals that the capillary effect reached a desirable level when the temperature exceeded 50 °C, indicating that the removal of hydrophobic impurities from the fiber required a certain amount of heat energy. Despite this, the required temperature was much lower than that of the conventional process. As bleaching at 70 °C was already found to provide a satisfactory *WI* and capillary effect, this temperature was adopted to investigate the effect of the duration.

#### 3.1.3. Effect of Duration

The fabrics were treated in the scouring and bleaching solutions containing 10 mmol/L SPC, 15 mmol/L TAED and 2 g/L penetrating agent at 70 °C for different durations (10–80 min). As revealed in Figure 4a, a prolonged duration led to a slightly increased *WI* and decreased *YI*, but a similar capillary effect. This implied that most colored and hydrophobic impurities were destroyed, dissolved and removed from the fiber in a relatively short time, e.g., 20 min. Such a short duration probably stemmed from the quick generation of PAA in the activated SPC system and efficient oxidation of colored impurities shown in Figure 1 [39]. Furthermore, extending the range of time further ensured the bleaching effect if necessary. The effects of the duration were less significant than those of the SPC concentration and temperature discussed above. As the duration of 30 min was already found to provide a satisfactory *WI* and capillary effect, this time was adopted in the following study.

#### 3.1.4. Comparisons of Scouring and Bleaching Effects

The conventional process was carried out according to Section 2.2. The activated SPC process was implemented at 70 °C for 30 min based on the above analyses. To comprehensively assess the performances of scouring and bleaching, multiple parameters, such as the *WI*, *YI* and capillary effect, were tested and are summarized in Table 1. 

Overall, it can be observed that the activated SPC system caused the fabric to have a higher *WI* and lower *YI* when compared with the conventional H_2_O_2_ system. For example, when 100 mmol/L 30% H_2_O_2_ was used, i.e., the same concentration of H_2_O_2_ contained in 10 mmol/L SPC, the *WI* of the activated bleached fabric (67.20) was higher than that of the conventional bleached fabric (64.59). Moreover, via the comparisons of pH value and capillary effect of the fabric, it was found that a near-neutral condition during the process and better hygroscopicity could be obtained by using the activated SPC system. 

#### 3.1.5. Energy Consumption Assessment

The theoretical energy consumption (*Q*) was approximately estimated based on the assumption that the width and length of the fabric were 1 m and 100 m, respectively, and the total weight of the fabric was 21 kg [7,34]. For the activated SPC process, *Q* was the total energy for heating the bleaching bath (*Q*_1_) and the treated fabric (*Q*_2_), and the dissipation of energy from the outside of equipment during the heat preservation stage (*Q*_3_). The mass ratio of the fabric to the bath was 1:25, and thus, the bleaching bath considered was 525 kg.
*Q*_1_ = *C*_1_*M*_1_ × (*T*_2_ − *T*_1_) = 1.296 kJ kg^−1^°C^−1^ × 21 kg × (70 − 25)°C = 1225 kJ
*Q*_2_ = *C*_2_*M*_2_ × (*T*_2_ − *T*_1_) = 4.186 kJ kg^−1^°C^−1^ × 525 kg × (70 − 25)°C = 98,894 kJ
*Q*_3_ = *αS*Δ*TD* = 0.02 kWm^−2^K^−1^ × 13.5 m^2^ × 25 °C × 0.5 h = 3.375 kW·h = 12,150 kJ
*Q* = *Q*_1_ + *Q*_2_ + *Q*_3_ = 112,269 kJ
where *T*_1_ and *T*_2_ are the temperatures of the environment and the bleaching bath, respectively; *C*_1_ and *C*_2_ are the specific heat capacity of cotton and water, respectively; *M*_1_ and *M*_2_ are the weights of fabric and bleaching bath, respectively; *α* is the heat transfer coefficient of air convection; *S* is the estimated surface area of equipment; Δ*T* is the temperature difference between the environment and outside of equipment; and *D* is the bleaching duration.

For a conditional H_2_O_2_ bleaching process, the theoretical energy consumption (*Q’*) was calculated according to the above method.
*Q’*_1_ = *C*_1_*M*_1_ × (*T’*_2_ − *T*_1_) = 1.296 kJ kg^−1^°C^−1^ × 21 kg × (95 − 25)°C = 1905 kJ
*Q’*_2_ = *C*_2_*M*_2_ × (*T’*_2_ − *T*_1_) = 4.186 kJ kg^−1^°C^−1^ × 525 kg × (95 − 25)°C = 153,836 kJ
*Q’*_3_ = *αS*Δ*T’D’* = 0.02 kWm^−2^K^−1^ × 13.5 m^2^ × 40 °C × 0.75 h = 8.1 kW·h = 29,160 kJ
*Q’* = *Q’*_1_ + *Q’*_2_ + *Q’*_3_ = 184,901 kJ

Unlike the parameters for the determination of Q, *T’*_2_ was the treatment temperature (95 °C). By contrasting *Q* and *Q*’, it can be found that the TAED/SPC activated process consumed 39.3% less energy than conventional bleaching. The evaluation result verified the sustainable advantage of SPC-activated bleaching in terms of energy saving. 

### 3.2. Bleaching Mechanism

In this section, the degradation of morin as one of the pigments in cotton was carried out in a homogeneous TAED/SPC system. To reveal the activated bleaching mechanism, the effects of the SPC concentration, temperature and radical scavenger on the retention rate of the morin (*C*/*C*_0_) were investigated. The structure of the degradation product was also analyzed to speculate the degradation pathway. 

As revealed in Figure 5a, the retention rate of the morin was about 70% with the absence of SPC, indicating that a small quantity of the morin was spontaneously decomposed under heating conditions at 70 °C. In the presence of SPC, *C*/*C*_0_ decreased dramatically from 70% to 10% after 30 min due to the oxidation of the morin by PAA. The decrease in pH value when adding SPC after 30 min also confirmed the production of PAA. It is worth noting that the degradation rate became slow when the SPC concentration (10 mmol/L) exceeded its stoichiometric amount relative to TAED. A mass of Na_2_CO_3_ released from the excess SPC probably led to the hydrolysis of TAED and the the invalid bimolecular decomposition of PAA [37,40]. Meanwhile, the excess H_2_O_2_ may consume some of the PAA via a nucleophilic attack [26]. These side reactions will result in the decrease of the concentration of PAA and reactive bleaching species, thereby decelerating the degradation rate. In general, the degradation ratio reached up to 90% and remained unchanged after 15 min in the TAED/SPC bath, which demonstrated the efficient capacity of the TAED/SPC system to degrade the pigment. 

As shown in Figure 5b, the degradation rate accelerated significantly with the increasing temperature from 25 °C to 90 °C. Figure 3a also showed that the *WI* of the fabrics gradually improved with the increasing temperature. The similar results verified the significant influence of temperature on pigment degradation shown in Section 3.1.2, i.e., the bleaching reaction rate in TAED/SPC system largely depended on the temperature. Moreover, the degradation ratio was about 95% after 10 min at 70 °C, which was close to the result at 95 °C. Thus, the effect of temperature on the morin degradation demonstrated that the TAED/SPC system functioned effectively at a low temperature, such as 70 °C. 

To explore the reactive oxygen species (ROS) that existed in the TAED/SPC system, radical scavengers, i.e., dimethyl sulphoxide (DMSO), benzoquinone (BQ) and sodium azide (NaN_3_), were separately added to the TAED/SPC bath. Figure 5c shows that the degradation ratio decreased from 97% to 24% after adding DMSO as the hydroxyl radical (HO·) scavenger. The strongly inhibited degradation reaction confirmed HO· as the main ROS that participated in the pigment degradation. It was reported that singlet oxygen (^1^O_2_) can be produced through the thermal decomposition of PAA [41]. To determine whether ^1^O_2_ plays a role in degradation, NaN_3_ as the scavenger of ^1^O_2_ was added to the TAED/SPC system. Figure 5c reveals an unobvious impact of NaN_3_ on the degradation ratio, indicating ^1^O_2_ was not the primary ROS responsible for degrading morin. This was consistent with the result of a previous study that showed that ^1^O_2_ existed in an o-phthalic anhydride-activated H_2_O_2_ system but had little effect on bleaching [32]. 

The superoxide radical (O_2_^−^·) is an effective bleaching ingredient during H_2_O_2_ bleaching, which can be generated via the pathway shown in Equations (3)–(5) [42,43]. Figure 5c shows that the degradation rate slowed down and the degradation ratio at 30 min decreased from 97% to 77% after adding BQ as the scavenger of O_2_^−^. The result demonstrated that O_2_^−^· was a contributor to the degradation of the morin. O_2_^−^· was further determined as a contributor using nitrotetrazolium blue chloride (NBT). Figure 5d displays the characteristic absorbance at 540 nm of the dark violet reaction solution, which demonstrated the reaction product of NBT with O_2_^−^· [44]. Thus, O_2_^−^· existed in the TAED/SPC system and mainly stemmed from the decomposition of SPC. Overall, both HO· and O_2_^−^· took part in the morin degradation, and the former ROS had a more obvious effect.
H_2_O_2_ + HO_2_^−^→HO^−^ + HO_2_· + HO·(3)
H_2_O_2_ + HO_2_^−^→H_2_O + O_2_^−^· + HO·(4)
HO_2_·→O_2_^−^· + H^+^(5)

To verify the actions of HO· and O_2_^−^· on the actual bleaching, the corresponding scavengers, i.e., DMSO and BQ, were incorporated into the TAED/SPC bleaching bath. As shown in Figure 5e, the *WI* decreased with the increasing concentration of DMSO or BQ, verifying that HO· and O_2_^−^· each played a role in the bleaching. A faster decreasing rate of the *WI* in the presence of DMSO signified a greater contribution of HO· in the bleaching. 

The bleaching mechanism of the TAED/SPC system was further researched via the analysis of the degradation path of the morin. Appendix A reveals the molecular ion peak at *m*/*z* 303 (M+1) in the LC-MS results of the undegraded morin with a retention time of 0.901 min. The LC-MS results of the morin degraded by TAED/SPC are shown in Appendix A. The disappeared molecular ion peak at *m*/*z* 303 demonstrated that the morin was completely degraded. The new molecular ion peaks were observed at *m*/*z* 145 (M+1) and *m*/*z* 167 (M+23), indicating the presence of DAED, which is the reaction byproduct of TAED and H_2_O_2_. To detect the degradation product of morin, DAED was separated using a column chromatogram. The purified degraded product was tested using MS and the result is illustrated in Figure 6. The new molecular ion peaks appeared at *m*/*z* 119, *m*/*z* 153, *m*/*z* 179, *m*/*z* 203, etc. It was speculated that the degradation products P_1_ and P_2_ were generated via a series of reactions, including (1) a radical reaction, (2,3) electron rearrangement, (4,6) an addition reaction and (5,7) a hydrolytic reaction. ^1^H-NMR was carried out to further testify the structures of P_1_ and P_2_. Appendix A clearly displays the hydrogen atoms in P_1_ and their chemical shifts of 6.26 (d, *J* = 2.4 Hz, 1H) and 6.24 (d, *J* = 2.4 Hz, 1H), as well as the hydrogen atoms in P_2_ and their chemical shifts of 7.87 (d, *J* = 8.8 Hz, 1H), 6.80 (d, *J* = 8.8 Hz, 1H) and 6.22 (s, 1H). 

### 3.3. Fabric Characterizations

#### 3.3.1. Mechanical Properties

The mechanical properties of the fabric before and after the pre-treatment are presented in Table 2. Compared with the greige fabric, the pre-treated fabric (samples b and c) exhibited a decreased breaking strength, but the breaking strength of the activated bleached fabric was higher than that of the traditionally bleached fabric, which demonstrated the advantage of the activated SPC system in diminishing fabric damage. The protection of the fiber from damage could be attributed to the mild pH condition provided by the activated SPC system shown in Table 1. 

#### 3.3.2. Dyeability with Reactive Dyes

The fabrics that were treated under different conditions exhibited differences in appearance and water absorbency (Table 1). A whiter appearance is required when a light-colored fabric is desired. Meanwhile, the hygroscopicity of the fabric plays a vital role in the adsorption and penetration of dyes during the initial stage of dyeing. Thus, both the activated bleached and conventional bleached fabrics were dyed in one bath to test whether the different *WI*s and capillary effects could cause a noticeable distinction in the dyeing properties between them. 

The dyeing property was evaluated by testing the color characteristics. Among them, chromatism (Δ*E*) over 3.0 means that there was an obvious color distinction between the two samples [45]. A higher *K/S* value indicates a deeper apparent color. As revealed in Table 3, when the dye dosage was only 0.5%, the same-colored fabrics displayed similar *K/S* values, but Δ*E* exceeded 3.0 when the red dye was used. By comparing the other samples, it was found that all Δ*E* values were close to or less than 3.0. These indicated that the differences in the *WI* and water absorbency had little impact on the dyeability when excluding the case of dyeing a light color. That is to say, the activated bleached fabric and conventional bleached fabric exhibited similar dyeability regarding reactive dyes.

#### 3.3.3. Micromorphology

The micromorphology of the cotton fabric before and after the pre-treatment was observed using SEM. Figure 7a,c show some white spherical impurities, including wax, cotton seed hull, oil and ash, on the surface of the raw fiber and conventionally treated fiber. The activated fiber (Figure 7b) maintained a similar morphology to that of raw cotton and conventionally treated cotton but showed a cleaner and smoother morphology than them. This phenomenon verified that the impurities on the fiber were completely removed by the activated SPC system and the fiber was effectively protected from damage. Additionally, some parallel ridges, obvious grooves and occasional breaks on the surface of raw fiber can be seen in Figure 7a. By contrast, the treated fabric surface became cleaner and flatter, indicating an obviously improved appearance after the scouring and bleaching.

#### 3.3.4. ATR FT-IR Spectra

ATR FT-IR spectroscopy as a surface-sensitive technique was performed to semi-quantitatively measure the presence of waxes on the cotton fabric. The fewer waxes that were found, the better the scouring effect that was obtained. In Figure 8, the spectrum of the greige (sample a) showed an extra peak at 2918 cm^−1^ corresponding to the asymmetric and symmetric stretching of methylene (-CH_2_-) groups in long alkyl chains [46,47]. This peak demonstrated the presence of waxes. Compared with sample a, sample c exhibited nearly the same absorbance intensity at 2918 cm^−1^, while sample b exhibited a weak absorbance intensity at 2918 cm^−1^. This fact signified that fewer waxes were detected for sample b. Thus, it could be considered that waxes were removed more adequately in the activated SPC system, indicating a better hygroscopicity of the fabric was obtained by using the activated SPC system. The result was consistent with the capillary effect shown in Table 1. Furthermore, the main characteristic absorption peaks of cotton cellulose were also observed using ATR FT-IR spectroscopy. Specifically, a broad band at 3408 cm^−1^ could be assigned to the free OH stretching vibration; absorption peaks at 1430 cm^−1^ and 1340 cm^−1^ could be assigned to the CH_2_ scissoring motion and C-H bending, respectively; and the bands at 1178 and 1049 cm^−1^ were dominated by C-O-C bond vibration and in-plane ring stretching, respectively [48,49]. A peak around 1640 cm^−1^ was due to the adsorbed water molecules [46].

## 4. Conclusions

The activated SPC system proposed in this study can be effectively applied in one-step low-temperature and near-neutral scouring and bleaching for cotton fabric. A satisfactory *WI* that increased by 67.6%, capillary effect (10.5 cm/30 min) and dyeing property was achieved after the fabric was treated in the optimized activated system that incorporated 10 mmol/L SPC, 15 mmol/L TAED and 2 g/L penetrating agent at 70 °C for 30 min. The temperature had a significant influence on both the *WI* and capillary effect, followed by the SPC concentration and the duration. An increased SPC resulted in an increased *WI*, but its concentration in excess of the stoichiometric amount relative to TAED could not produce an obvious additional effect. Compared with the conventional H_2_O_2_ system, which was utilized at 95 °C for 45 min, the SPC-activated system exhibited the advantages of lower energy consumption (energy savings of approximately 39.3%), lower fiber damage (<5%) and milder process conditions (pH 7.5–8.5). The SEM revealed that the SPC-activated bleached fiber showed a smoother and cleaner surface. The ATR FT-IR indicated that waxes could be removed more adequately by the activated SPC system. In terms of the bleaching mechanism, the reactive species that participated in degrading the cotton pigment were the hydroxyl radical and superoxide radical based on the results of the radical scavenger experiment, and the contribution degree of the former was larger than that of the latter. Two degradation components with molecular weights of 180 and 154 were detected using mass spectroscopy. According to the above analyses, the bleaching mechanism of the TAED/SPC system, which included a radical reaction, electron rearrangement, an addition reaction and a hydrolytic reaction, was proposed. Overall, TAED/SPC is a promising bleaching system that is beneficial to the low-carbon sustainable development of the textile industry.

## Figures and Tables

**Figure 1 materials-15-05849-f001:**
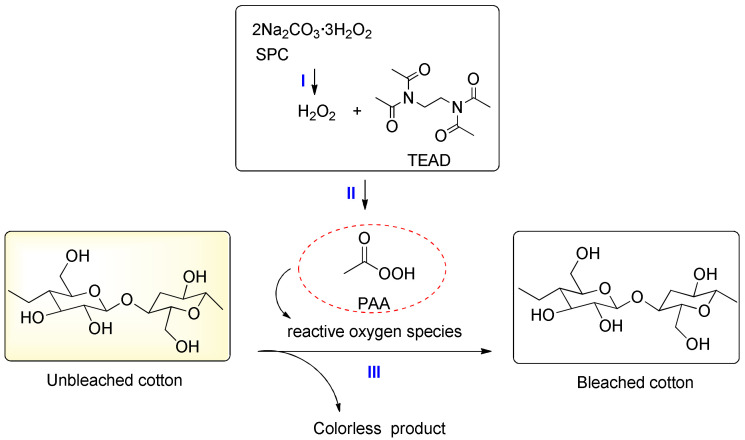
Bleaching mechanism of a TAED/SPC-activated system for the cotton fabric.

**Figure 2 materials-15-05849-f002:**
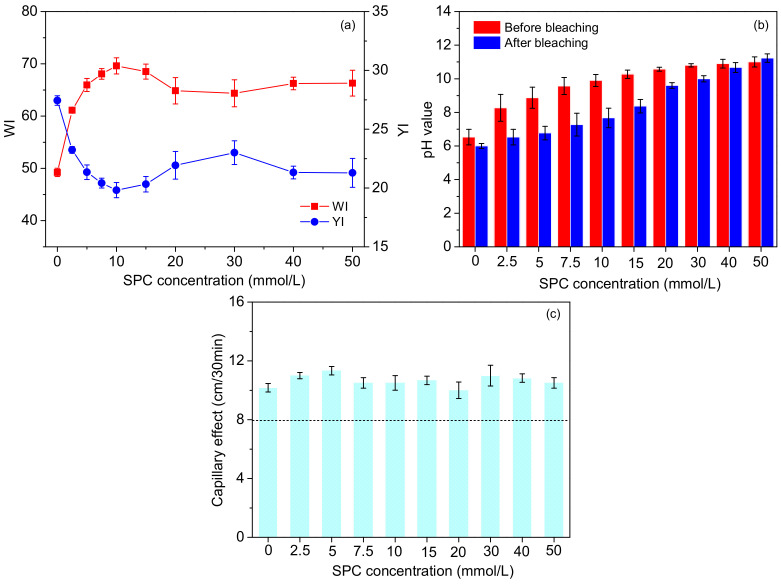
Effects of the SPC concentration on the *WI/YI* (**a**) and capillary effect (**c**) of the fabrics; pH variations of the processing solutions before and after bleaching (**b**).

**Figure 3 materials-15-05849-f003:**
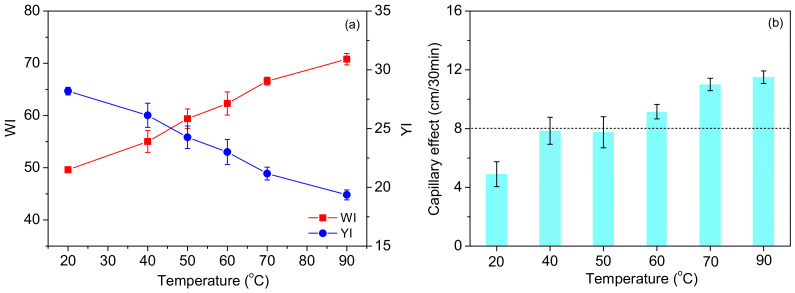
Effects of the temperature on the *WI/YI* (**a**) and capillary effect (**b**) of the fabrics.

**Figure 4 materials-15-05849-f004:**
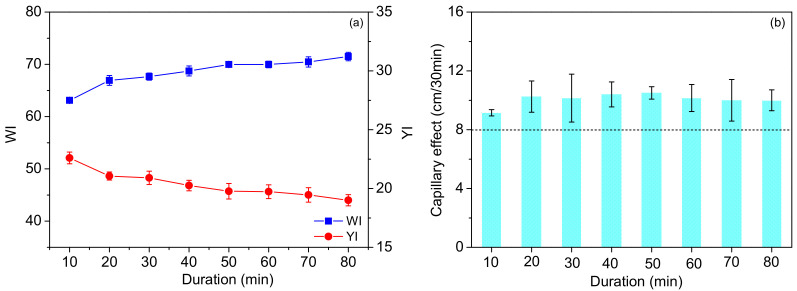
Effects of the duration on the *WI/YI* (**a**) and capillary effect (**b**) of the fabrics.

**Figure 5 materials-15-05849-f005:**
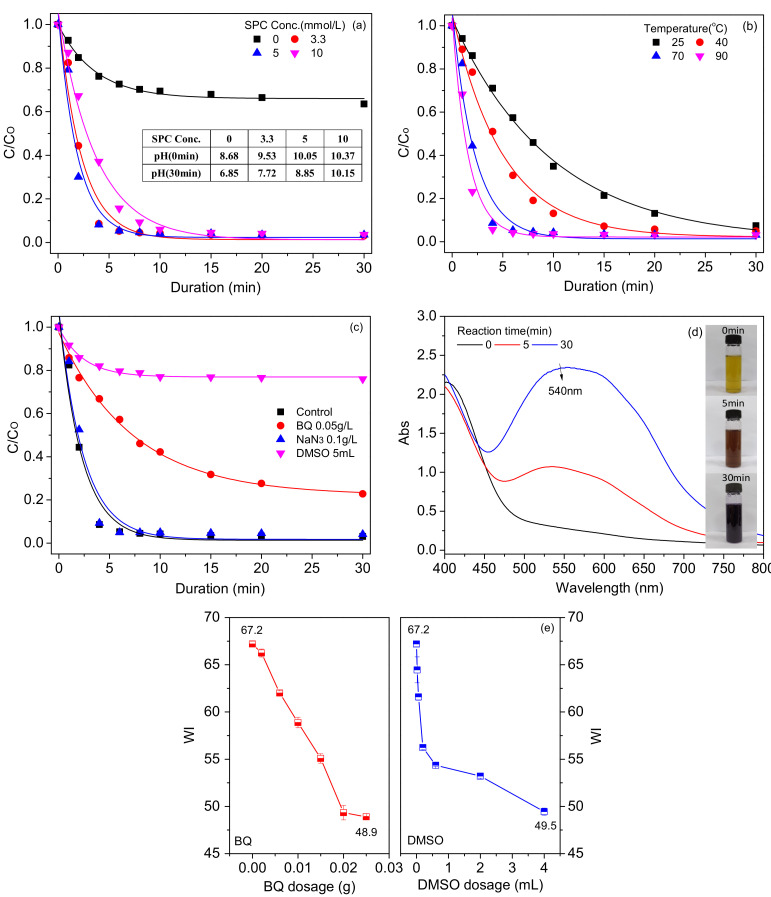
Effect of the (**a**) SPC concentration and (**b**) temperature on the degradation of the morin with the TAED/SPC system; (**c**) influences of DMSO, BQ and NaN_3_ on the morin degradation in the homogeneous TAED/SPC system; (**d**) UV-Vis spectra of the NBT that reacted with the TAED/SPC; (**e**) effect of the concentration of DMSO and BQ on the *WI* of the cotton fabric bleached by the TAED/SPC system. Conditions: morin 120 μmol/L (**a**–**c**), NaOH 1 mmol/L (**a**–**c**), TAED 5 mmol/L (**a**–**d**), 70 °C (**a**,**c**,**d**), SPC 3.3 mmol/L (**b**–**d**) and NBT 0.2 g/L (**d**).

**Figure 6 materials-15-05849-f006:**
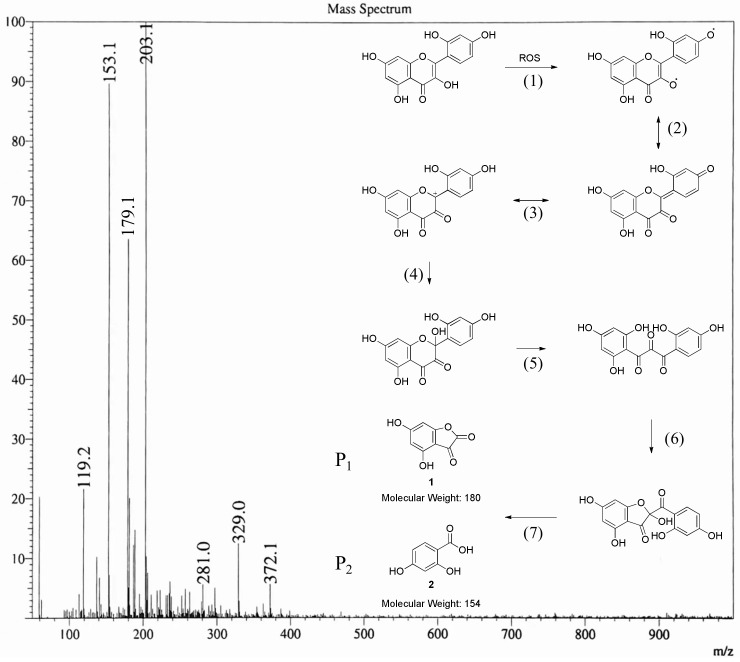
MS of degradation product (negative ion mode) and speculated degradation pathway of the morin.

**Figure 7 materials-15-05849-f007:**
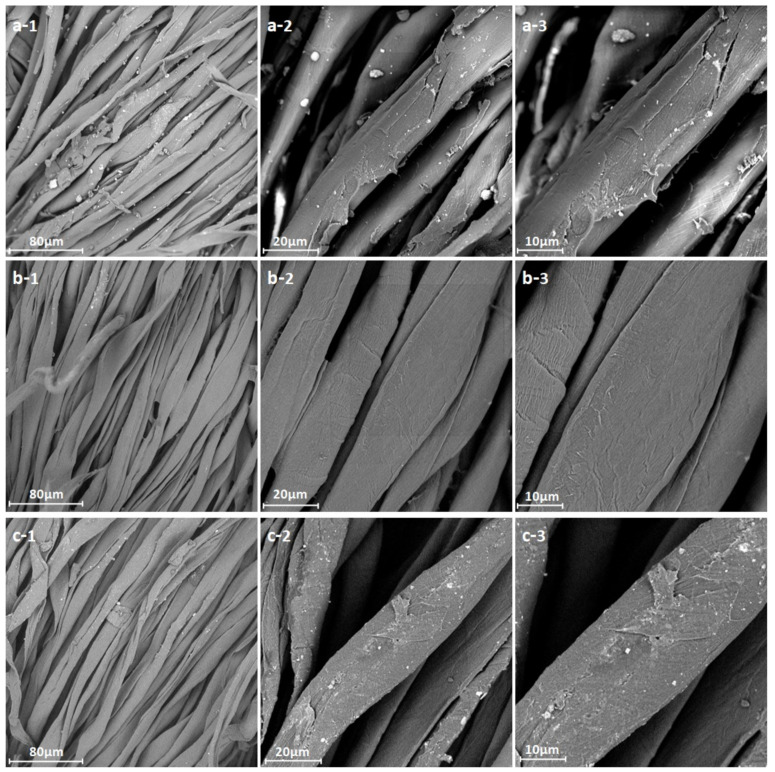
SEM images of the greige and bleached fabric: (**a**) greige, (**b**) fabric bleached using the SPC-activated system and (**c**) fabric bleached using a conventional system; 1, 2 and 3 stand for the magnifications of 1000, 3000 and 5000, respectively.

**Figure 8 materials-15-05849-f008:**
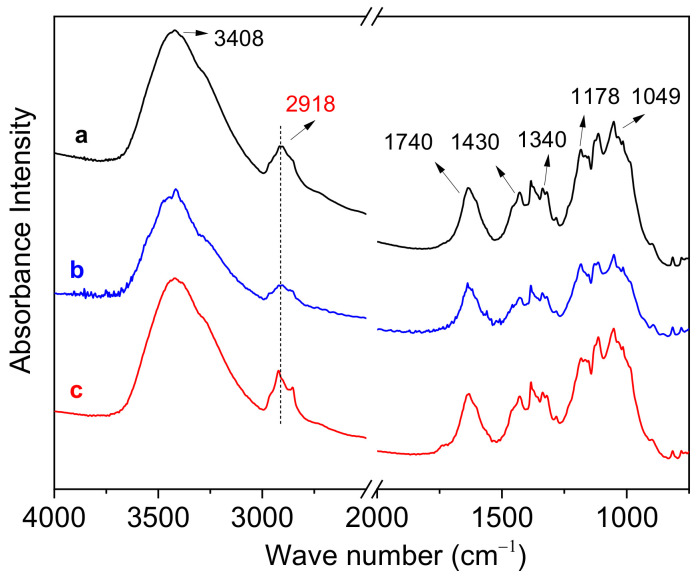
ATR FT-IR spectra of the greige and bleached fabric: (**a**) greige, (**b**) fabric bleached using the SPC-activated system and (**c**) fabric bleached using a conventional system.

**Table 1 materials-15-05849-t001:** Comparisons of the scouring and bleaching performances between the two systems.

Sample	30% H_2_O_2_ (mmol/L)	SPC (mmol/L)	TAED (mmol/L)	*WI*	*YI*	pH ^I^	pH ^II^	Capillary Effect (cm/30 min)
Greige				40.10	32.23	-	-	0.5
1	0	2.5	3.75	55.19	25.81	9.0	7.5	10.2
2	0	10	15	67.20	20.86	9.8	8.0	10.5
3	0	27	40	71.29	19.00	10.5	8.5	10.3
4	0	50	75	72.11	18.80	10.8	8.5	9.4
5	30	0	0	55.31	25.58	11.0	10.5	1.1
6	100	0	0	64.59	21.98	11.0	10.7	2.4
7	150	0	0	68.32	20.23	10.7	10.7	1.2
8	300	0	0	69.11	20.04	10.7	10.7	2.2
9	400	0	0	71.19	19.15	10.5	10.5	2.3
10	800	0	0	73.54	18.27	10.5	10.5	4.5

^I^ before bleaching; ^II^ after bleaching.

**Table 2 materials-15-05849-t002:** Comparisons of the mechanical properties of the fabric.

Sample	Breaking Strength (N)		Elongation at Break (%)	
	Warp	Weft	Warp	Weft
a	595 ± 44	531 ± 30	26.84 ± 2.58	25.77 ± 2.22
b	573 ± 26	529 ± 22	33.28 ± 1.96	31.16 ± 1.72
c	515 ± 21	509 ± 24	34.14 ± 2.07	29.31 ± 1.84

^a^ greige; ^b^ fabric bleached using the SPC-activated system; ^c^ fabric bleached using a conventional system.

**Table 3 materials-15-05849-t003:** Color characteristics of the dyed cotton fabric.

Dye Used	Dye Dosage (o.w.f %)	Sample	*L*	*a*	*b*	*K/S*	Δ*E* *
Red	0.5	b	58.28	50.28	−7.8	2.99	3.99
		c	60.31	46.89	−8.35	2.35	
	1.0	b	50.90	55.87	−5.95	5.99	1.90
		c	51.87	54.38	−6.63	5.23	
	2.0	b	44.51	58.02	−2.65	10.81	1.72
		c	45.78	57.43	−3.65	9.42	
Yellow	0.5	b	82.24	12.51	54.05	2.06	2.07
		c	81.95	13.87	55.58	2.21	
	1.0	b	78.52	18.87	65.51	4.17	2.05
		c	77.85	20.80	65.36	4.30	
	2.0	b	74.44	25.67	73.00	7.55	3.04
		c	75.77	24.75	70.43	6.13	
Blue	0.5	b	48.48	−8.04	−20.43	4.04	1.82
		c	46.70	−7.86	−20.74	4.59	
	1.0	b	40.35	−7.21	−21.32	7.29	0.63
		c	40.82	−7.17	−20.90	6.91	
	2.0	b	31.27	−5.44	−20.20	13.71	0.74
		c	30.59	−5.22	−20.01	14.30	

^b^ fabric bleached using the SPC-activated system; ^c^ fabric bleached using a conventional system; * Δ*E* is the color difference between dyed fabric b and dyed fabric c.

## Data Availability

All data that support the findings of this study are available from the corresponding authors upon reasonable request.

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
