# Peer review of "Energy-Saving One-Step Pre-Treatment Using an Activated Sodium Percarbonate System and Its Bleaching Mechanism for Cotton Fabric"

_materials, 2022, doi:10.3390/ma15175849_

Round 1

Reviewer 1 Report

The article from Li and coworkers presents a one-step pre-treatment process by activated sodium percarbonate and its bleaching mechanism for cotton fabric.

The work is scientifically sounding, but it:

1. requires a major revision before resubmission (specific comments are provided in the following)

2. The approach adopted to assess the effects of the input variables (T, SPC concentration) would be more effective by using suitable statistical design of the experiment and analyisis of data. It is not evident, at the end of the work, a clear conclusion on the effect supported by a robust statistical analysis

Finally, the paper and the research here presented probably better fits a more specialized journal dedicated to fabric processing.

Specific comments

Line 45. Check English

Line 66. COD - Every time you introduce a ne acronym, please provide a complete definition

Line 77. Idem for TAED. Even if you introduced the acronym in the abstract, you should repeat in the main text. tetraacetylethylenediamine (TAED)

Line 89. Please provide the correct defintion using the chemical formula of SPC. Avoid common language.

Line 117. revealed?

Line 124-126. actually, the work from https://doi.org/10.1111/cote.12474 econmpasses some mechanisms.

Line 203. FZ is a chinese standard. It is difficult to access for most of the readers of the paper. Please provide a brief description.

Line 208. Plesae provide the entire title of the standards

Line 225. OPUS software? provide specifications

Section 3.1.1 Effect of SPC concentration.

In this section it is not reported the dataset related to the untreated cotton. Data of the pristine material is useful to better understand the starting point.

Please provide data of pristine cotton, that can be useful for th following sections.

Moreover, a comparison to conventional treatements that you reported in section data easily available from literature) can be useful as benchmark.

Section 3.3.1 Mechanical property. Table 2. Please report data error (i.e. standard deviation). The discussion of the mechanical results require for a statistical analysis. Differences seem negligble and no clear conclusions can be extrapolated from.

line 446. Revise English

3.3.2 Dyeability with reactive dyes. It is not evident to me which is the base materials that you are comparing calculating Delta E. please better clarify

3.3.4 ATR FI-IR spectra. Did you try to quantitavively assess the peaks indicating the presence of wax?

Reviewer 2 Report

The authors presented an activated pre-treatment method for cotton fabric which uses sodium percarbonate. They have compared the tradition and the activated method regarding several parameters. However, the text is not clear enough to be easily understood. The authors must improve the language of the manuscript before it can be published anywhere.

The text needs English improvement.

The abstract is very confusing and needs improvement.

Various passages of the manuscript have more a textbook character and lengthily explain fundamental aspects that have been reported before.

In Section 2.6.1. Colour features, the authors presented equations (1) and (2) but did not reference them. Could the authors reference the equations and discuss them?

In section 3.1.1. the authors presented the pH variation after bleaching. The increase in pH probably difficult the decomposition of PAA. Why? And why does the pH increase after bleaching?

Why do increasing temperature gives a direct energy to accelerate the release of H2O2 from SPC?

The authors said that WI of the activated bleached fabric, of 67.20, was much higher than that of conventional bleached fabric, 64.59. Why is 67.20 “much higher” than 64.59? This seems to me a subjective and unsupported analysis.

Do the authors have a hypothesis on why the activated bleached fabric and conventional bleached fabric exhibited a similar dyeability towards to reactive dyes?

Reviewer 3 Report

In this manuscript, a low-temperature, near-neutral system of bleaching was designed using tetraacetyleth-16 ylenediamine (TAED) activated sodium percarbonate (SPC) system for the bleaching of cotton fabric in order to reduce waste and high energy consumption by the conventional system of bleaching. The manuscript was well written except the obvious grammatical infractions, however, the results presented is interesting and provides a new insight to ways by which bleaching can be carried out faster in an environmentally friendly manner. I recommend the publication of this manuscript in Material after the authors have considered the comments below.  

Comments.

1.      “Traditional pre-treatment of cotton fabric produces abundant energy consumption and wastewater discharge which is hardly meets the requirement of low carbon emissions”. The language of the sentence above is grammatically incorrect, kindly correct. More so, the entire manuscript is replete with grammatical errors, I suggest the authors read through thoroughly and seek for help if need be

2.      What is the advantage of using high concentration of sodium percarbonate over the mild hydrogen peroxide used for the conventional bleaching of cotton fabrics?

3.      A decrease in the breaking strength of activated SPC bleached fabric cannot be advantage, however, the authors discussed it as advantage when it wasn’t compared to the conventionally bleached fabric but rather to the greige fabric. I suggest the authors compare breaking strength visa vis the conventional bleaching systems as against activated SPC.

4.      From the colorimetric study, what is the whiteness index of activated SPC bleached fabric visa vis the conventionally bleached fabric?

. No comparative studies have been made in terms of energy savings for activated SPC bleached fabric and the conventional bleaching. The energy savings in your title is only suggestive. I recommend you presents a studies on that.

Round 2

Reviewer 2 Report

The authors have addressed all the questions raised by this reviewer.